# Treatment of Pregnant Women with Ivermectin during Mass Drug Distribution: Time to Investigate Its Safety and Potential Benefits

**DOI:** 10.3390/pathogens10121588

**Published:** 2021-12-08

**Authors:** Astrid Christine Erber, Esther Ariyo, Piero Olliaro, Patricia Nicolas, Carlos Chaccour, Robert Colebunders

**Affiliations:** 1Department of Epidemiology, Center for Public Health, Medical University of Vienna, 1090 Vienna, Austria; astrid.erber@meduniwien.ac.at; 2Centre for Tropical Medicine and Global Health, Nuffield Department of Medicine, University of Oxford, Oxford OX3 7FZ, UK; piero.olliaro@ndm.ox.ac.uk; 3Center for Population Family Health, University of Antwerp, 2600 Antwerp, Belgium; Esther.Ariyo@uantwerpen.be; 4Instituto de Salut Global Barcelona, Hospital Clínic-Universitat de Barcelona, 08036 Barcelona, Spain; patricia.nicolas@isglobal.org (P.N.); carlos.chaccour@isglobal.org (C.C.); 5Facultad de Medicina, Universidad de Navarra, 31008 Pamplona, Spain; 6Centro de Investigación Biomédica en Red de Enfermedades Infecciosas, 28029 Madrid, Spain; 7Global Health Institute, University of Antwerp, 2610 Antwerp, Belgium

**Keywords:** onchocerciasis, ivermectin, pregnancy, safety, epilepsy, nodding syndrome, trial, registry

## Abstract

To date, pregnant women are excluded from programmes delivering community-directed treatment of ivermectin (CDTI) for onchocerciasis and preventive chemotherapy of other helminthiases because of concerns over ivermectin safety during pregnancy. This systematic exclusion sustains an infection reservoir at the community level and deprives a vulnerable population from known benefits—there are indications that treating *O. volvulus* infected women may improve pregnancy outcomes and reduce the risk that their children develop onchocerciasis-associated morbidities. Furthermore, teratogenic effects are seen in non-clinical experiments at doses that far exceed those used in CDTI. Lastly, early, undetected and undeclared pregnancies are being systematically exposed to ivermectin in practice. Treatment of this population requires appropriate supporting evidence, for which we propose a three-pronged approach. First, to develop a roadmap defining the key steps needed to obtain regulatory clearance for the safe and effective use of ivermectin in all pregnant women who need it. Second, to conduct a randomised placebo-controlled double-blind clinical trial to evaluate the safety and benefits of ivermectin treatment in *O. volvulus* infected pregnant women. Such a trial should evaluate the possible effects of ivermectin in reducing adverse pregnancy outcomes and neonatal mortality, as well as in reducing the incidence of onchocerciasis-associated epilepsy. Third, to establish a pregnancy registry for women who inadvertently received ivermectin during pregnancy. This situation is not unique to ivermectin. Access to valuable therapies is often limited, delayed, or denied to pregnant women due to a lack of evidence. Concerns over protecting vulnerable people may result in harming them. We need to find acceptable ways to build robust evidence towards providing essential interventions during pregnancy.

## 1. Introduction

Onchocerciasis impacts heavily on poor and vulnerable populations in remote areas of sub-Saharan Africa. This disease causes severe disability, including blindness and epilepsy, resulting in significant psycho-social and economic consequences. Currently, onchocerciasis elimination programmes rely mainly on annual or bi-annual community-directed treatment with ivermectin (CDTI) [1]. Using CDTI, the African Programme for Onchocerciasis Control (APOC) was successful in eliminating onchocerciasis as a public health problem in several African countries [1]. However, in many onchocerciasis-endemic areas in Africa, such as in conflict-stricken zones that experienced periods of insecurity and where CDTI coverage has been sub-optimal or interrupted, onchocerciasis transmission remains high with some areas experiencing a high prevalence of onchocerciasis-associated epilepsy (OAE) [2,3]. Moreover, preliminary data suggest a possible neurocognitive decline in onchocerciasis-exposed children, even without, or prior to, the onset of epilepsy [4].

Pregnancy is currently considered as a contra-indication for ivermectin treatment during CDTI even though formal pregnancy testing is not carried out before the distribution of ivermectin [1]. Women are only told not to take ivermectin if they may be pregnant based on their last menstrual period [5]. However, women are often reluctant to reveal their pregnancy status to ivermectin distributors due to uncertainty, social risk, and privacy considerations. This is particularly true during the early stages of pregnancy, and even at later stages in the case of unplanned or unwanted pregnancies. Pregnant women have thus been potentially exposed to inadvertent ivermectin treatment at scale.

About 3.7 billion doses of ivermectin have been distributed in mass drug administration (MDA) campaigns globally over the past 30 years [6]. In a study in Ghana, during an MDA of ivermectin and albendazole, 14.6% of pregnant women were inadvertently treated. This represented 1.7% of women of childbearing age (15−49 years) [7]. It has been estimated that up to 50% of women in the first trimester of pregnancy in onchocerciasis endemic areas of Africa may have received ivermectin [8,9].

## 2. What Is Known about the Safety of Ivermectin during Pregnancy?

Currently it is not indicated to use ivermectin during pregnancy because insufficient safety data are available concerning its use during pregnancy in humans [9]. The developmental toxicity of ivermectin was investigated in mice, rats, rabbits and dogs. The results demonstrated that teratogenic effects (cleft palates in mice, rats and rabbits, and clubbed fore-feet without skeletal alterations in rabbits) were produced only at doses similar to those causing severe maternal toxicity. The no-observed-effect level (NOEL) for teratogenicity in the most sensitive species and strain, the CF-1 mouse, was 0.2 mg/kg body weight, while the NOEL for maternal toxicity was 0.1 mg/kg body weight. A sub-population of the CF-1 mice used in these studies was later found to be deficient in P-glycoprotein expression, an efflux pump with a central role in preventing ivermectin toxicity [6,10,11,12,13,14]. Teratogenicity of ivermectin in animal studies was observed at cumulative doses ranging between 20 and 600 times the human single-dose target of 150–200 μg/kg given during CDTI [5]. Ivermectin has been widely used in veterinary medicine (in pigs, cattle, horses, dogs, and sheep) for many years. Although the number of pregnant animals treated is unknown, the incidence of adverse reproductive effects following ivermectin in treatment is extremely low in all species [14].

A recent systematic review and meta-analysis on the safety of ivermectin treatment in pregnant women [6] identified 496 pregnant women (500 pregnancy outcomes) who inadvertently received ivermectin during mass drug administration (MDA) campaigns, and 397 (399 pregnancy outcomes) treated as part of an open-label randomised clinical trial (RCT) carried out in Masindi, Uganda [15]. Based on these studies, no statistically significant difference in pregnancy outcomes between ivermectin-treated pregnant women and concurrent control groups was observed. There were no reported neonatal deaths, maternal morbidity, preterm births, or low birthweight infants reported in any included study [6]. In an RCT in Masindi, Uganda, comparing the efficacy of ivermectin and 400 mg albendazole alone or combined for treating soil-transmitted helminth infections in the second trimester of pregnancy, the abortion rate was lower than recorded in earlier reports [7,8]. This might be ascribed to the anti-parasitic effect of treatment or, as the authors suggested, to the exclusion of women with a history of habitual abortion and enrolment of women in their second trimester of pregnancy, as most abortions occur in the first trimester.

In conclusion, there is a lack of safety data about the use of ivermectin during pregnancy, but there is also no evidence that ivermectin during pregnancy is teratogenic in humans. Therefore, given the potential important benefits of ivermectin treatment of pregnant women, such a treatment should be considered despite potential risks. Following this argument, the French national reference centre for teratogenic agents of France no longer recommends excluding pregnant women with helminthiases from ivermectin treatment [16]. There is thus a precedent for prescribing ivermectin during pregnancy, provided an appropriate risk-balance assessment is conducted.

## 3. What Is Known about the Negative Impact of Onchocerciasis in Pregnancy

*Onchocerca volvulus* can be transmitted in utero to the developing foetus and prime the immune response in newborns [17,18,19,20,21]. The severity of onchocerciasis and its associated itching is an independent predictor of a shorter lactation period [22]. An *O. volvulus* infection in pregnant women may be associated with an increased risk of spontaneous abortions [23] and with earlier and more severe *O. volvulus* infections in their offspring [24].

## 4. The Effect of Ivermectin Treatment of *O. volvulus* Infected Pregnant Women on Pregnancy Outcomes

Ivermectin treatment of *O. volvulus* infected pregnant women may improve pregnancy outcomes. Indeed, a study in Ecuador suggested that ivermectin treatment in an onchocerciasis-endemic area decreased the frequency of spontaneous abortions [20]. The study compared the incidence rates of spontaneous abortions between onchocerciasis-endemic and non-endemic areas. Between 1982 and 1989, a marked increase in spontaneous abortions of 9.5/1000 maternal years at risk (myar) was observed in the onchocerciasis-endemic area and was associated with an increase in community *O. volvulus* microfilarial (mf) load, potentially related to the El Niño climatic phenomenon, which resulted in a marked increase in rainfall and the blackfly population [25]. In the same period, the incidence rate of spontaneous abortions in the non-endemic area was 1.3/1000 myar. An MDA with ivermectin every six months started in 1990 in the hyper-endemic area and 90−95% of those eligible were treated [23]. This resulted in a dramatic decline in community mf load to negligible levels by 1996 [23]. This large decrease in community mf load was accompanied by a decrease in spontaneous abortion rates (5.0/1000 myar) so that, by 1992, they were equivalent to those in the non-endemic area (2.3/1000 myar) [23].

## 5. An *O. volvulus* Infection in a Pregnant Women Induces Parasite Tolerance in Their Offspring

It is possible that ivermectin treatment of *O. volvulus* pregnant women with ivermectin may reduce the risk of their children developing onchocerciasis-associated-morbidities. The reason for this is parasite tolerance. Indeed, the intra-uterine exposure to filarial antigens reduces cellular responses to parasite antigens in their offspring [26] and increases the risk of post-natal filarial infection [26,27,28]. Maternal *O. volvulus*-infection will sensitize in utero parasite-specific cellular immune responsiveness in neonates and activate *O. volvulus* Ag-specific production of several Th1- and Th2-type cytokines [21]. Immune responses with a preferential Th2 pattern induce parasite tolerance in newborns [29] and neonatally induced specific immune responses will persist upon secondary antigen contact later in life [30].

In an 18-year follow-up study of approximately 4000 families in West Africa, maternal *O. volvulus* infection was associated with a four-fold increase in *O. volvulus* infection risk in children [24]. Moreover, in the same study children born to *O. volvulus* infected mothers had an increased risk of becoming infected earlier in life and developing a higher persistent mf load [24]. In two cohort studies in Cameroon, a high mf load has been shown to be a risk factor for developing epilepsy [31,32].

Recent epidemiological studies suggest that onchocerciasis triggers the development of onchocerciasis-associated epilepsy (OAE) and other neurological disabilities [30]. OAE appears in previously healthy children between the ages of 3 and 18 years with a peak age of onset around 8–11 years [33]. This contrasts with non-endemic regions in Africa where most epilepsy in children develops before the age of 5 years and is due to perinatal causes and genetic antecedents. OAE presents with a wide spectrum of seizures, including generalized tonic-clonic, nodding seizures (nodding syndrome), and stunting and delayed puberty (Nakalanga syndrome) [33].

In a recent case–control study in northern Uganda, we found that pre-term birth could be a risk factor for nodding syndrome (Gumisiriza, manuscript under review). We hypothesize that pre-term birth could be related to an untreated *O. volvulus* infection during pregnancy. In an onchocerciasis-endemic area in Cameroon, we determined the presence of onchocerciasis antibodies in 209 school-age children without epilepsy and assessed their neurocognitive performance using a battery of validated tools. Applying multiple logistic regression analysis and adjusting for age, gender, education level, previous ivermectin use, and BMI-for-age z-scores, the presence of *O. volvulus* antibodies was significantly associated with reduced semantic verbal fluency (OR: 0.91 [0.84–0.99]) and lower scores on the International HIV Dementia Scale (OR: 0.85 [0.73–0.98]) [4]. Furthermore, past ivermectin use was associated with increased neurocognitive scores. These findings suggest that exposure to *O. volvulus* may affect the neurocognitive performance of children [4]. Moreover, a study in Uganda suggested that nodding syndrome may be preceded by prodromal features (excessive sleepiness, slowing down of activities, decline in comprehension, blank staring) over a period of a few weeks to two years [34].

Parasite tolerance in children born to infected pregnant mothers has also been observed with other filarial infections [35]. However, a study in India revealed that although a proportion of infected mothers had cleared their lymphatic filariasis (LF) infection due to ongoing MDA, the children born to them acquired the infection at comparable rates to the children born to mothers who had not cleared the infection [35]. It is therefore possible that ivermectin treatment of *O. volvulus* in pregnant women may not decrease the risk of their children developing OAE because parasite tolerance may also be transferred by all mothers who have been exposed to the parasite. Therefore, in addition to including pregnant women with an active *O. volvulus* infection in a trial to assess the effect of ivermectin on pregnancy outcomes, pregnant women with past *O. volvulus* infection only presenting OV16 antibodies and pregnant women without infection also need to be followed.

## 6. Potential Other Advantages of Treating *O. volvulus* Infected Pregnant Women

Parasitic infections in pregnant women may lead to reduced responses to vaccinations. A study in Kenya reported that filaria-tolerant infants (born to LF-infected mothers) showed a reduced response to the *Haemophilus influenza* vaccine [36]. In another Kenyan study, mother-to-child transmission of HIV was significantly higher in women co-infected with one or more helminths (48%) compared to women without helminthic infections (10%; adjusted OR, 7.3; 95% CI, 2.4–33.7) [37].

*Strongyloides* hyper-infection may cause both maternal and neonatal death [38] and it has also been suggested that strongyloidiasis during pregnancy may impact the cognitive and motor functions of their children [39,40]. As ivermectin is the treatment of choice for strongyloidiasis, it has been suggested that ivermectin should be considered in *Strongyloides*-infected pregnant women [41].

Currently, only topically applied scabicide drugs are available for treating scabies infestation in pregnant women. Ivermectin, a very effective drug to treat scabies, has been proposed as second-line drug in scabies-infested pregnant women after the end of organogenesis (e.g., 10 weeks of amenorrhea). Indeed, in cases of scabies with eczematous or superinfected skin with risk of sepsis, oral ivermectin is the only alternative [41].

## 7. *O. volvulus* Infected Pregnant Women Constitute a Parasitic Reservoir

Not treating pregnant women may delay onchocerciasis elimination goals. In a study in Nigeria, 35% of the 1714 women of reproductive age who presented during the CDTI round in the year 2000 were excluded because they were pregnant or nursing babies. Only 56% of them were treated with ivermectin later in the year [42]. Particularly in remote communities in South Sudan, it is very difficult to reach the CDTI coverage threshold required for onchocerciasis elimination (>80%) if pregnant or suspected pregnant women are excluded (Logora M, NTD Programme Director). Although ivermectin is very effective in killing microfilaria, it is less effective in decreasing the fertility of the female worm [39]. Given the fact that *O. volvulus* transmission is still ongoing in certain onchocerciasis foci in Africa, despite many years of CDTI, there is growing concern about the possibility that ivermectin resistance may develop [43]. There is evidence of a loss of sensitivity of *O. volvulus* to ivermectin from genome-wide analyses and epidemiological studies [44,45,46]. A population of women at child-bearing age that is only intermittently treated with ivermectin because of consecutive pregnancies could increase this risk of developing resistance.

## 8. The Way Forward

All this calls for a concerted effort, agreed upon by relevant stakeholders in disease-endemic countries and internationally, to build a robust evidence base for the use of ivermectin in pregnancy. We propose a three-pronged approach to achieve this:

1. To design a roadmap which will define the key steps needed to obtain regulatory clearance for the safe and effective use of ivermectin in all pregnant women who need it.

2. To undertake a randomised placebo-controlled double-blind clinical trial to determine the safety and efficacy of ivermectin treatment in *O. volvulus* infected pregnant women, examining potential improvements in pregnancy outcomes while also identifying whether ivermectin protects children from developing sequelae such as OAE. Such a trial should be carried out in onchocerciasis-endemic regions with high ongoing *O. volvulus* transmission and a high prevalence of OAE such as in parts of South Sudan, Cameroon and the Democratic Republic of Congo. Conducting such a trial in these remote areas will be challenging because in these areas where onchocerciasis-elimination programmes have been performing sub-optimally, the health care infrastructure is also weak, and establishment of an appropriate research infrastructure with qualified personnel to carry out such a trial will require experience and will be challenging, but immensely beneficial to the population in the future. This trial should include, at its planning stage, qualitative research conducted in cooperation with the local communities and focusing on gender and vulnerability issues, in order to establish an appropriate and acceptable recruitment process, informed consent procedure and trial conduct.

3. To establish an international pregnancy registry to record pregnancy outcomes in women who were exposed to ivermectin during their pregnancy inadvertently or through an informed consent procedure.

## 9. Conclusions

To date, in MDA programmes with ivermectin, albendazole and praziquantel, pregnant women are excluded because the potential teratogenic effects of these anti-parasitic drugs are not known. However, in practice, when these drugs are delivered as preventive chemotherapy during mass administration and community-delivered programmes, pregnancy tests are not done, which means many pregnant women are systematically inadvertently exposed to these drugs.

This situation is not unique to ivermectin or antiparasitic drugs at large in disease-endemic countries. It is a general issue. Like women who are not pregnant, pregnant women need to use drugs to manage chronic or acute conditions, but to the extent that there is labeling information for pregnant women, it is often based on nonclinical data with or without limited human safety data [47]. Access to valuable therapies is often limited, delayed, or denied to pregnant women due to concerns over their safety, or potential teratogenicity. Concerns over protecting vulnerable people may result in harming them. At the same time, accidental exposures are often ignored or under-investigated. In line with current FDA recommendations [47], we need to find acceptable ways to build robust evidence, through well conducted clinical trials and pregnancy registries, towards providing essential interventions during pregnancy. This is particularly true for low- and middle-income countries with high birth rates, where women might be excluded from treatment, or be at risk from inadequate treatments, for extended periods of time during their reproductive years, thus exacerbating their vulnerability.

## Data Availability

Not applicable.

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
