# Peer review of "Treatment of Pregnant Women with Ivermectin during Mass Drug Distribution: Time to Investigate Its Safety and Potential Benefits"

_pathogens, 2021, doi:10.3390/pathogens10121588_

Round 1

Reviewer 1 Report

It is an interesting review paper about a very common issue found on the clinical practice.

Results came from a literature review. Data have clearly expressed, and redaction make an easy reading. Congratulations

Author Response

Reviewer 1

It is an interesting review paper about a very common issue found on the clinical practice.

Results came from a literature review. Data have clearly expressed, and redaction make an easy reading. Congratulations

Response

Thanks

Reviewer 2 Report

Erber A.C. and colleagues : “Treatment of pregnant women with ivermectin during mass drug distribution: time to investigate its safety and potential benefits.”

Filarial diseases are a major issue in humans. The manuscript raises the relevant question of the pregnant women infected with Onchocerca volvulus who are excluded from the MDA programmes with ivermectin and thus contribute to the parasitic reservoir. The logic of the risk-balance assessment is sounded. Overall, the manuscript is well written and organized. This is an interesting opinion not subjected to overinterpretations that merits publication and will definitely benefit to the readers of Pathogens. However, a number of points should be addressed as detailed below.

Need to indicate the dosing rates administered in every studies reported. This is very important information when comparing the potential impact of ivermectin during pregnancy, in pregnancy outcome, women offspring, etc. (lines 81-87, 105-116,etc.)

The manuscript should mention toxicological evidence from the use of ivermectin in animal health in relation with animal gestation, fetal abortion, newborn health and filarial parasites (dog, cattle,..). What lessons can be drawn from the Onchocerca ochengi and Onchocerca gutturosa models ?

The authors should tell about the emergence of resistance to ivermectin in Onchocerca volvulus reported in the literature in the face of the problematic raised in the manuscript.

Section5: the section title does not correspond to the content of the section which is mostly related to parasite tolerance and not to ivermectin treatment. Please change the title.

Add some relevant references missing and comment :

Prichard RK, Basa´n˜ ez MG, Boatin BA, McCarthy JS Garcı´a HH, et al. (2012) A Research Agenda for Helminth Diseases of Humans: Intervention for Control and Elimination. PLoS Negl Trop Dis 6(4): e1549. doi:10.1371/journal.pntd.0001549

El-Saber Batiha G, Alqahtani A, Ilesanmi OB, Saati AA, El-Mleeh A, Hetta HF, Magdy Beshbishy A. Avermectin Derivatives, Pharmacokinetics, Therapeutic and Toxic Dosages, Mechanism of Action, and Their Biological Effects. Pharmaceuticals (Basel). 2020 Aug 17;13(8):196. doi: 10.3390/ph13080196.

Line 52-54: please indicate any reference illustrating this point

Line 77: please indicate what RCT means

Line 111: please indicate « microfilarial (mf) load »

Line 133, 205, 209, : Italics for O. volvulus

Lines 132-160: these lines are not related to the effect of ivermectin treatment which it the main focus of section 5. To be moved into section 3 an adjust the section title.

Lines 198-222, Section8: It is suggested a 3-spronged approach but only 2 points are appearing. Please modify.
